# ZO-1 Regulates Hippo-Independent YAP Activity and Cell Proliferation via a GEF-H1- and TBK1-Regulated Signalling Network

**DOI:** 10.3390/cells13070640

**Published:** 2024-04-05

**Authors:** Alexis J. Haas, Mert Karakus, Ceniz Zihni, Maria S. Balda, Karl Matter

**Affiliations:** UCL Institute of Ophthalmology, University College London, London EC1V 9EL, UK; alexis.haas@gmail.com (A.J.H.); m.karakus@ucl.ac.uk (M.K.); c.zihni@liverpool.ac.uk (C.Z.)

**Keywords:** tight junctions, transcription, YAP, GEF-H1, TBK1, ZO-3, ZONAB, cytoskeleton, myosin, focal adhesions

## Abstract

Tight junctions are a barrier-forming cell–cell adhesion complex and have been proposed to regulate cell proliferation. However, the underlying mechanisms are not well understood. Here, we used cells deficient in the junction scaffold ZO-1 alone or together with its paralog ZO-2, which disrupts the junctional barrier. We found that ZO-1 knockout increased cell proliferation, induced loss of cell density-dependent proliferation control, and promoted apoptosis and necrosis. These phenotypes were enhanced by double ZO-1/ZO-2 knockout. Increased proliferation was dependent on two transcriptional regulators: YAP and ZONAB. ZO-1 knockout stimulated YAP nuclear translocation and activity without changes in Hippo-dependent phosphorylation. Knockout promoted TANK-binding kinase 1 (TBK1) activation and increased expression of the RhoA activator GEF-H1. Knockdown of ZO-3, another paralog interacting with ZO1, was sufficient to induce GEF-H1 expression and YAP activity. GEF-H1, TBK1, and mechanotransduction at focal adhesions were found to cooperate to activate YAP/TEAD in ZO-1-deficient cells. Thus, ZO-1 controled cell proliferation and Hippo-independent YAP activity by activating a GEF-H1- and TBK1-regulated mechanosensitive signalling network.

## 1. Introduction

Epithelial and endothelial tissue development and homeostasis require the control of cell proliferation and death. Cell–cell adhesion contributes to the regulatory pathways that guide these processes, but the underlying molecular mechanisms are still incompletely understood [1,2,3]. Similarly, cell and tissue mechanics impact cell adhesion, cell proliferation and death, but if and how such regulatory pathways are linked is not clear.

Tight junctions (TJs) are part of the epithelial and endothelial junctional complex, and are required for barrier formation [3]. They have also been linked to the regulation of transcription and cell proliferation in vitro and in vivo; however, the underlying molecular mechanisms are still incompletely understood [3,4,5,6].

ZO-1 is a central TJ scaffolding protein that engages in multiple protein–protein interactions and regulates TJ formation in a tension-dependent manner that is modulated by the mechanical properties of the ECM [7,8]. ZO-1 depletion induces cell-wide changes in cytoskeletal organization, increased focal adhesion formation, and traction on the ECM [8]. Analysis of different epithelial cancers suggests that ZO-1 may play an important role in tumourigenesis and metastasis [9]. Indeed, knockout of ZO-1 in the normally non-proliferative adult retinal pigment epithelium stimulated G1/S phase transition and cell death [10]. Global knockout also resulted in massive cell and embryonic death [11]. However, relevant downstream effectors are not well understood. Overexpression data indicate that ZO-1 inhibits G1/S phase cell cycle progression by cytosolic sequestration of the transcriptional and posttranscriptional regulator ZONAB [4,5]. In contrast, recent loss of function experiments questioned whether knockdown/knockout of ZO-1 alone or combined with reduced ZO-2 expression indeed stimulates proliferation, as no effects on G1/S phase transition were observed after serum starvation [12].

Regulation of the transcriptional activity of the YAP/TAZ pathway is a key mechanism in the control of tissue proliferation and development. A central regulator of YAP/TAZ activity is the Hippo pathway; however, alternative Hippo-independent processes also impact YAP/TAZ activity, such as cytoskeletal tension and actomyosin activity [13]. In epithelia, the Hippo pathway is regulated by adherens junctions and the apical polarity machinery, and guides YAP/TAZ signalling by phosphorylation [14,15,16]. The junctional adhesion protein JAM-A has been linked to the regulation of the Hippo pathway in intestinal cells [17]. Epithelial YAP/TAZ signalling is regulated by the cell density, resulting in the downregulation of YAP/TAZ activity and cytosolic sequestration with increasing cell density, a process that is at least in part regulated by E-cadherin engagement and mechanical strain [18]. The TJ protein ZO-2 has also been investigated for its contributions to YAP/TAZ regulation. However, the reported results are contradictory and, even if activated, YAP did not stimulate increased proliferation in ZO-2-depleted cells [12,19]. Despite the importance of TJs and ZO-1 in the regulation of actomyosin and cytoskeletal tension, no effects on YAP/TAZ-mediated transcription by ZO-1 have been reported. Similarly, it is not clear if TJs can regulate YAP/TAZ by Hippo-independent mechanisms.

ZO-1 forms mutually exclusive complexes with either ZO-2 or ZO-3 [20]. ZO-3 is a 130 kD protein [21,22]. Expression of a ZO-3 construct containing N-terminal domains has been shown to affect total cellular RhoA activity [23]. Unlike ZO-1, ZO-3 has been proposed to be required for proliferation due to the stabilization of cyclin D1 protein expression [24]. However, ZO-3 has not been linked to the regulation of transcription.

We have recently established transient depletion and stable knockout approaches to study the roles of ZO-1, revealing, unexpectedly, distinct functions in the regulation of epithelial morphology and junction assembly [8]. We now used these cell models to investigate the role of ZO-1 in the regulation of epithelial proliferation. Given ZO-1 depletion stimulated basal actomyosin activity and, consequently, increased cell–ECM traction [8], we also asked whether ZO-1 regulates Hippo-independent YAP/TAZ activity, a transcriptional regulatory mechanism regulated by ECM stiffness and basal mechanotransduction. Our data indicate that ZO-1 is required for the normal control of cell proliferation, and that ZO-1 depletion/knockout stimulates proliferation by driving Hippo-independent YAP activation and cell proliferation via a regulatory network that requires GEF-H1, TANK-binding kinase (TBK1), and mechanotransduction at focal adhesions.

## 2. Materials and Methods

### 2.1. Cell Culture

Experiments with MDCK II knockout cells were performed with cell lines described previously, using two ZO-1 knockout clones (MDCK ZO-1KO C1 and C2), a ZO-1/ZO-2 double knockout clone (MDCK ZO-1/2KO), and the corresponding wild-type MDCK II cell line (MDCK wt) [8,25]. siRNA-mediated knockdown of ZO-1 was performed in an MDCK II strain that has been previously described [8,26]. The GFP-mZO-1 MDCK cell line was generated using a mouse ZO-1 cDNA [8]. Small molecule inhibitors were purchased from Tocris Bioscience (Abingdon, UK) and diluted as 1000× stocks in DMSO: MRT68601 (2 mM); blebbistatin (10 mM), CK666 (20 mM), SMIFH2 (25 mM stock), and Y27632 (10 mM). Hydrogels were prepared as described [7,27]. Preparation of Matrigel-coated polyacrylamide hydrogels and glass coverslips was performed as described previously [7,27]. Briefly, glass coverslips (22, 13, and 10 mm diameter round coverslips) were washed with 70% ethanol, dried at 70 °C, and then exposed to UV light overnight. The 22 and 10 mm coverslips were then coated with Matrigel by incubation with a solution of 65 μg/mL Matrigel for 2.5 h at 37 °C. The 13 mm coverslips were silanized for 3 min using ethanol containing 0.37% Bind-Silane solution (GE Healthcare Life Science, Gillingham, UK) and 3.2% acetic acid. The protocol of Tse and colleagues was used to prepare polyacrylamide and bis-acrylamide (N,N′-methylenebisacrylamide) solutions corresponding to elasticities of 1 kPa and 40 kPa [28].

### 2.2. Experimental Setups

For siRNA experiments, cells were seeded at 3 × 10^5^ cells/well into 6-well plates and transfected with siRNAs the following day. After 24 h, the cells were trypsinized, resuspended in full medium, and seeded in experimental plates containing the required substrates. 7 × 10^3^ cells/well were plated in 48-well plates containing glass coverslips, and 24-well plates containing 40 kPa or 1 kPa hydrogels were seeded with 12.5 × 10^3^ cells/well or 25 × 10^3^ cells/well, respectively. For immunoblotting, cells were plated accordingly using plates of different sizes with or without hydrogels. Experimental plates were processed after 48 h of culture. For experiments with knockout clones, a corresponding two-step method was used to facilitate comparison (i.e., seeding at 3 × 10^5^ cells/well into 6-well plates for 48 h, before reseeding on different substrates at the same concentrations as for the RNA interference experiments. Experimental plates were also processed after 48 h of culture. For proliferation assays, 500 cells were plated into wells of 96-well plates and cells were cultured for up to 11 days. For experiments with Transwell filters (0.4 μm diameter pores), 3 × 10^5^ cells were seeded per filter and cultured for six days. Media were replaced every second day in proliferation assays and filter cultures.

### 2.3. siRNA Transfections

RNAiMAX transfection reagent (ThermoFischer Scientific, Walham, MA, USA) was used for siRNA transfections [8]. Briefly, cells were transfected in 6-well plates using 8 μL/well RNAiMAX and 8 μL/well of 20 μM siRNA. In double knockdown experiments, siRNAs against two targets were mixed at a 1:1 ratio and, in samples targeting only one mRNA, siRNAs against the single target were mixed at a 1:1 ratio with control siRNA. The following canine siRNAs were used: ZO-1: 5′-CCAUAGUAAUUUCAGAUGU-3′ and 5′-CCAGAAUCUCGAAAAAGUGCC-3′; Talin: 5′-GUCCUCCAGCAGCAGUAUAA-3′ and 5′-GAGGCAACCACAGAACACAUA-3′; GEF-H1 5′-AGACACAGGACGAGGCUUA-3′, 5′-GGGAAAAGGAGAAGAUGAA-3′, and 5′-GUGCGGAGCGGAUGCGCGUAA-3′; TBK1: 5′-GACAGAAGUUGUGAUCACA-3′ and 5′-CUCUGAGUACCAUAGGAUU-3′; YAP: 5′-GACUCAGGAUGGAGAAAUA-3′, 5′-GGAGAGGAGUUGAUGCCAA-3′ and 5′-GCAGGAACUUUGCCCGAAA-3′; TAZ: 5′- GACUUCCUUAGCAACGUGG -3′, 5′- CCCCAAGCCCAGCUCGUGG-3′ and 5′-GGCUUCAGAGGAUCCAGAU-3′; ZONAB: 5′-AGACGUGGUUACUAUGGCA-3′ and 5′-CAACGUCAGAAAUGGAUAU- 3′; and ZO-3: 5′- GGGAGGAGGCUGUGCAGUU-3′ and 5′-GCGUGAUCGCGGAGAAGAA-3′.

### 2.4. Reporter Gene Assays

Plasmids containing a promoter with binding sites for specific transcription factors to be measured, driving firefly luciferase expression, were co-transfected with a renilla luciferase reporter plasmid with a CMV promoter (Promega Corp, Madison, WI, USA) using TransIT reagent (0.25 μL/well, 96-well plate, and 1 μL/well, 48-well plate, Mirus Bio, Madison, WI, USA). The transfection mix was replaced with fresh medium after 4 h. The following day, the two luciferases were measured sequentially using the dual luciferase assay kit (Promega Corp, Madison, WI, USA) and a BMG FLUOstar OPTIMA microplate reader (Ortenbert, Germany). The TEAD-reporter plasmid was kindly provided by Stefano Piccolo [29]; the TCF-reporter (TOPFLASH) and a corresponding control plasmid (FOPFLASH) were purchased from Upstate Biotechnology Inc. (Lake Placid, NY, USA); the cyclin D1 promoter plasmid and the ZONAB reporter plasmid were as described in [4,30].

### 2.5. Antibodies and Immunological Methods

Methods for immunofluorescence and immunoblotting were previously described [7,31]. The following antibodies were used: YAP, mouse monoclonal (12395, Lot 3, Cell Signaling, Danvers, MA, USA); phoslpho-Ser127 YAP, rabbit polyclonal (4911, Lot 5, Cell Signaling); TAZ, rabbit polyclonal (sc-48805, Santa Cruz Biotechnology, Dallas, TX, USA); MLC2, rabbit polyconal (3672, Lot 6, Cell Signaling); pMLC2, rabbit polyconal (6671, Lot 6, Cell Signaling, Danvers, MA, USA) and mouse monoclonal (3675, Lot 6, Cell Signaling); ppMLC2, rabbit polyconal (95777, Lot 1, Cell Signaling); Ki67, mouse monoclonal (18-0192Z, Lot 789595A, ThermoFisher Scientific, Walham, MA, USA); occludin, mouse monoclonal (33-1500, Lot TC259714, ThermoFisher Scientific, Walham, MA, USA); TBK1, rabbit polyclonal (3013, Lot 4, Cell Signaling); phospho-Ser172 TBK1, rabbit polyclonal (5483, Lot 15, Cell Signaling); cingulin, rabbit polyclonal (ab244406, Lot GR326501-29, abcam); and talin, mouse monoclonal (T3287, Lot 49M4782V, Sigma-Aldrich, St. Louis, MO, USA). GEF-H1 was detected with rabbit polyclonal anti-peptide antibodies generated against N- and C-terminal peptides [32]. For immunoblotting for ZO-1, -2, and -3, rabbit polyclonal anti-peptide antibodies were used [32]. ZONAB was also detected with rabbit polyclonal antibodies against N- and C-terminal peptides [4]. A mouse monoclonal antibody against α-tubulin was previously described [33]. The following affinity-purified and cross-adsorbed secondary antibodies from Jackson ImmunoResearch (Ely, UK) were used: Alexa488-labelled donkey anti-rabbit (711-545-152) and anti-mouse IgG (115-545-003), Cy3-labelled donkey anti-rabbit (711-165-152) and anti-mouse IgG (715-165-150), and Cy5-labelled donkey anti-goat IgG (705-175-147) were diluted 1/400 from 50% glycerol stocks. Phalloidin-Atto647 (65906) and Hoechst-33342 were from Sigma-Aldrich. For immunoblotting, affinity-purified HRP-conjugated goat anti-mouse (115-035-003) and anti-rabbit (111-035-144) secondary antibodies were purchased form Jackson ImmunoResearch and diluted 1/5000 from 50% glycerol stocks, and affinity-purified IRDye 800CW donkey anti-mouse (926-32212) and anti-rabbit (926-32213), and 680LT donkey anti-mouse (926-68072) and anti-rabbit (926-68023) secondary antibodies from LI-COR and diluted 1/10000. Enhanced chemiluminescence (X-ray film or a Bio-Rad ChemiDoc, Bio-Rad Laboratories Hercules, CA, USA) or fluorescence (Li-Cor Odyssey, Li-Cor, Lincoln, NE, USA) were used for detection [8]. The gel densitometry toolset in ImageJ/Fiji (Version 2.9.0) or Adobe Photoshop (Version 2023, Adobe, San Jose, CA, USA) was used to quantify immunoblot images.

### 2.6. Cell Proliferation, Necrosis, and Apoptosis Assays

For cell number assays, cells plated in 96-well plates were removed from the incubator at the days indicated in the experiments and, after aspirating the medium, frozen at −20 °C until analysis but for at least 18 h. After defrosting, the CyQuant cell proliferation assay kit (ThermoFisher Scientific, Walham, MA, USA) was used to measure the cell numbers using 4-times the standard concentration of the fluorescent dye to avoid saturation of the assay. Necrosis was quantified by measuring release of lactate dehydrogenase in 30 μL samples of the medium and apoptosis by determining cell-associated caspase3/7 activity using the respective kits from Promega Corp. (Madison, WI, USA); CytoTox-ONE homogeneous membrane integrity assay and Apo-ONE homogeneous caspase-3/7 assay) [34]. All three assays were performed with the same cell samples, starting with the LDH secretion assay followed by the caspase-3/7 assay and, after freezing and thawing the plates, the cell number assay.

### 2.7. Light Microscopy

A Nikon Eclipse Ti-E epifluorescence microscope with a CFI Apochromat Nano-Crystal 60× oil objective (N.A., 1.2) (Nikon Europe, Amstelveen, The Netherlands) or a Leica TCS SP8 with an HC PL APO 40× (N.A., 1.30) or 63× (N.A., 1.40) oil objectives (Leica Microsystems, Wetzlar, Germany) were used for immunofluorescence imaging.

### 2.8. Image Processing and Quantifications

Microscopy and immunoblot images were processed and adjusted using ImageJ/Fiji (Version 2.9.0) and Adobe Photoshop CC software (Version 2023, Adobe, San Jose, CA, USA). Quantifications by measuring specific integrated densities were performed with ImageJ/Fiji. Quantification of YAP nuclear translocation in ZO-1 siRNA experiments was performed using a pipeline of ImageJ/Fiji macros and a Python script. Briefly, cells grown on glass or polyacrylamide hydrogels were immunostained and imaged by acquiring multichannel z-stack images using the Nikon Eclipse Ti-E epifluorescence microscope with the 60× oil objective. The most in-focus image of the nuclei staining channel, the YAP staining channel, and the F-actin staining channel, respectively, were automatically selected with the “Find focused slices” plugin (Qingzong Tseng; https://sites.google.com/site/qingzongtseng/find-focus, accessed on 20 October 2022). A 2D segmentation of the cell nuclei and cell bodies (outlined by the lateral F-actin staining) was performed using Cellpose’s pretrained model [35]. Label images of nuclei were converted into ROIs using the “Label to ROIs” function from the MorpholibJ plugin [36] to create a mask that was used to delete nuclei areas from the labels of the cell bodies. The label image of nuclei and the newly created label image of cell bodies were then converted back into ROIs using the “pullLabelsToROIManager” function form the CLIJ2 library [37]. YAP fluorescence intensity was then measured on the YAP staining channel using the ROIs of the nuclei and the ROIs of the cell bodies. Nuclear and cytosolic intensity measurements were label-matched using a Python script to obtain a final ratio of nuclear translocation for single cells. In knockout cell lines and ZO-2 depletion experiments, nuclear YAP localization was quantified with ImageJ by measuring the specific fluorescence intensity of equal nuclear and cytoplasmic areas. Ki67-positive cells, mitotic indexes, and nuclear layers were obtained by manual counting.

### 2.9. Statistics and Reproducibility

For reporter gene assays, cell survivability assays, apoptosis assay, LDH-necrosis assay, and immunoblots, the n numbers are reflected by the provided data points in graphs that refer to the number of independent experimental repetitions. For quantifications of microscopy images, data points reflect either images or cells analysed as specified in the figure legends. No samples were measured repeatedly. Statistical significance was tested using Kruskal–Wallis and Wilcoxon tests, or ANOVA and two-tailed *t*-tests. For pairwise multiple comparisons, Tukey HSD and Steel–Dwass tests were used. All quantifications show all data points along with median and interquartile ranges, or means and standard deviations. Graphs and statistical calculations were generated with JMP-Pro (Version 16) or GraphPad Prism (Version 9).

## 3. Results

### 3.1. ZO-1 Regulates Cell Proliferation and Death

We first asked whether knockout of ZO-1 in MDCK cells impacts cell proliferation. In all knockout clones, we measured increased cell counts when cells were plated at low density and left to proliferate over 11 days (Figure 1A–C). Quantifications at day 6, when wild-type cells had reached about 50% of their final cell density, revealed that single ZO-1 knockout cells proliferated faster than ZO-1/2 double knockout cells. This was paralleled by increased LDH secretion—a measure for necrosis—and caspase 3/7 activity—a measure for apoptosis (cell death markers were divided by cell numbers to correct for increased cell counts) (Figure 1D). As expected from the increase in proliferation, activity of the cyclin D1 promoter also increased in knockout cells (Figure 1E). However, a TCF-regulated promoter was not affected by ZO-1 knockout, indicating that β-catenin signalling, a pathway regulated by adherens junctions, was not altered (Figure 1F). Thus, ZO-1 knockout in MDCK cells stimulated proliferation and increased apoptosis and necrosis. The ZO-1/2 double knockout cells exhibited an even larger increase in necrosis, which might be the reason for the slower increase in cell numbers and the larger proliferation drop towards the end of the experiments when cells reached confluency (Figure 1C,D).

We next asked whether cells continued to proliferate once they reached high densities. Hence, cells were plated on filters to ensure efficient medium access. The cells were fixed after 6 days, a time by which wild-type and knockout clones developed stable barrier properties [8]. Staining for the proliferation marker Ki67 revealed that, unlike wild-type cells, the ZO-1 knockout cells continued to proliferate as about half the cells remained Ki67-positive (Figure 1G,H). Similarly, quantification of mitotic indices revealed a fourfold increase in ZO-1 KO cells (Figure 1I). Thus, ZO-1 knockout cells continued to proliferate at high cell densities, indicating loss of cell density-dependent proliferation control. Hence, a balance between proliferation and cell death/extrusion likely enables the cells to form confluent cell layers. As we have shown previously, this balance between proliferation and death does not disable cells to form functional barriers as ZO-1 knockout does not prevent the formation of functional tight junctions although they are structurally deficient and have altered permeability properties [8].

Confocal sectioning revealed that the monolayer structure was distorted with areas that appeared multilayered (Figure 1G). However, the cells remained normally polarized with distinct staining for ezrin, an apical marker, and scribble, a basolateral polarity protein (Appendix A). Highly confluent cells on filters reflect a monolayer under low tension as the cells are very compact; hence, we tested whether monolayer deformation was mechanosensitive by repeating the confocal analysis with cells plated on ECM of different stiffnesses that had formed large islands but were still subconfluent [8]. Strikingly, cell layer deformation was not observed in large islands of subconfluent cultures on stiff ECM but was pronounced on 1 kPa ECM (Appendix A). As many cells were very extended and could be traced across the entire layer of cells, it is likely that the structures were not formed by the multilayering of cells but primarily by morphological deformation, resulting in stretching and loss of the regular predominantly perpendicular orientation relative to the substrate. Thus, ZO-1 knockout resulted in increased cell proliferation and death, and a mechanosensitive deregulation of cell morphology and monolayer architecture.

### 3.2. ZO-1 Regulates YAP/TEAD Activity

Depletion of ZO-1 induces cell-wide changes in cytoskeletal tension [8]. Hence, we asked whether ZO-1 depletion influenced YAP/TAZ signalling [13]. Knockout of ZO-1 alone or together with ZO-2 indeed stimulated the YAP/TAZ-activated transcription factor binding element TEAD activity twofold (Figure 2A). YAP is phosphorylated by Hippo pathway activation, leading to cytosolic YAP retention [13]. Hence, we used an antibody that allows monitoring changes in YAP phosphorylation at serine 127, a Hippo pathway target site [38]. YAP phosphorylation at serine-127 was not affected (Figure 2B,C). Thus, enhanced TEAD activity in knockout clones was unrelated to regulation by the Hippo pathway.

A previous RNA interference study suggested that ZO-1 depletion does not stimulate YAP [12]. Hence, we performed knockdown experiments using previously described siRNAs along with a cell line expressing GFP-tagged mouse ZO-1, which is resistant to the herein used canine siRNAs (Figure 2D) [8]. As in knockout cell lines, depletion of ZO-1 resulted in a strong increase in TEAD-responsive promoter activity (Figure 2E). Stimulation was prevented by expression of mouse ZO-1, supporting the specificity of the depletion-induced stimulation and the used siRNAs (Figure 2E). In agreement with the knockout experiments, depletion of ZO-1 resulted in a strong increase in nuclear YAP (Figure 2F). Increased nuclear localization was observed when individual or pooled siRNAs were used and was inhibited by the expression of an siRNA-resistant to GFP-tagged mouse ZO-1 (GFP-mZO-1; Figure 2G). Thus, depletion of ZO-1 stimulated nuclear YAP accumulation and increased TEAD activity.

As YAP is a mechanosensitive transcriptional co-factor [29], we asked if decreasing monolayer tension affects the ZO-1 depletion-induced increase in nuclear YAP localization by seeding cells on Matrigel-coated elastic substrates with tuneable stiffness [8]. When cells were seeded on stiff ECM (40 kPa), ZO-1 depletion still stimulated increased nuclear YAP localization. However, on soft 1 kPa hydrogels, much of the detected YAP remained cytosolic (Appendix A), suggesting that ECM stiffness modulates the effect of ZO-1 on YAP localization. Similarly, measurements of TEAD activity indicated that ZO-1 depletion stimulated TEAD activity on stiff and soft matrices but that the absolute activity was lower on 1 kPa than on 40 kPa ECM (Appendix A). Thus, ZO-1 depletion leads to increased TEAD transcriptional activity and increased YAP localization in a manner that is modulated by ECM stiffness.

### 3.3. TBK1 Is Required for Regulation of YAP by ZO-1

The serine/threonine kinase TANK-binding kinase 1 (TBK1) can regulate YAP in a Hippo-independent manner [39]. Immunoblotting for TBK1 phosphorylated at serine-172 indicated an increase in phosphorylated TBK1 in ZO-1 and double knockout cell lines (Figure 3A,B). Serine-172 resides in the activation loop; hence, increased phosphorylation indicates increased activity.

We next asked whether TBK1 was involved in the control of ZO-1 over YAP localization and activity. We first used immunofluorescence to determine the effects of TBK1 inhibition on YAP nuclear accumulation. This revealed that the inhibition of TBK1 strongly attenuated the nuclear accumulation of YAP in ZO-1-deficient as well as double knockout cells (Figure 3C,D). Inhibition of TBK1 also inhibited TEAD-responsive promoter activity (Figure 3E). TBK1 siRNAs, which efficiently depleted expression of the kinase in MDCK cells (Figure 3F), also suppressed the reporter assay for TEAD activity (Figure 3G), confirming the inhibitor data. Inhibition of the TEAD reporter assay upon TBK1 knockdown was also observed in MDCK cells depleted of ZO-1 using siRNAs (Appendix A).

Therefore, the regulation of YAP nuclear localization and activation of TEAD-regulated transcription by ZO-1 relies on a TBK1-dependent mechanism that requires catalytic activity of the kinase.

### 3.4. TBK1 Depletion Promotes Junction Formation by ZO-1-Deficient Cells

ZO-1 depletion inhibits TJ formation, which can be rescued by knockdown of GEF-H1 [8]. Hence, we tested whether TBK1 also cooperates with GEF-H1 during TJ formation on a stiff substrate that induces high junctional tension [7].

As observed previously, depletion or knockout of ZO-1 led to a disrupted TJ phenotype based on staining with anti-occludin antibodies in cells that had not been grown to equilibrium density on stiff matrices that result in increased cytoskeletal tension on cell junctions [8]. Transfection of TBK1 siRNAs on their own did not affect TJ formation (Figure 4A,B; see Appendix A for immunoblots). Cο-transfection of ZO-1 and TBK1 siRNAs or transfection of ZO-1 KO cells with TBK1 siRNAs stimulated TJ formation by ZO-1-deficient cells (Figure 4). Thus, TBK1 signalling contributes to the defect in TJ formation induced by ZO-1 deficiency.

### 3.5. ZO-1 Regulates YAP/TAZ Activity in a GEF-H1- and Focal Adhesion-Dependent Manner

The RhoA exchange factor GEF-H1 is released from TJs upon ZO-1 ablation [8]. GEF-H1 is inactive at TJs due to an inhibitory interaction with cingulin and, when released, stimulates RhoA signalling [40]. GEF-H1 has been proposed to interact with TBK1 to enhance its signalling responses [41]. Hence, GEF-H1 may cooperate with TBK1 in ZO-1-deficient cells to stimulate YAP signalling.

We first measured expression of GEF-H1 by immunoblotting using a polyclonal anti-peptide antibody [32]. Quantification revealed a more-than-twofold increase in GEF-H1 expression in ZO-1 KO cells (Figure 5A,B). siRNA-mediated depletion of ZO-1 also stimulated increased GEF-H1 expression (Appendix A).

We next asked if GEF-H1 was involved in the regulation of YAP by ZO-1. RNAi-mediated depletion of GEF-H1 strongly inhibited the induction of TEAD transcriptional activity in the ZO knockout clones (Figure 5C,D). GEF-H1 is thought to stimulate actomyosin contractility. Inhibiting actomyosin contractility with the myosin II inhibitor blebbistatin or inhibiting actin polymerization by the Arp2/3 complex with CK-666 prevented TEAD activity induction (Figure 5E). These results thus indicate that GEF-H1 and actomyosin contractility were required for YAP stimulation in ZO-1 knockout cells.

GEF-H1 is thought to promote actomyosin activity along the basal membrane in epithelia by inducing stress fibres [40]. ZO-1 knockdown stimulates focal adhesion formation and ECM traction [8]. Hence, we asked whether inhibiting basal mechanotransduction in cells grown on glass alters the induction of YAP activity observed upon ZO-1 depletion. Talin functions at the interface of integrins and the actin cytoskeleton in focal adhesions [42,43]. It is essential for mechanotransduction at focal adhesions. The depletion of talin in ZO-1 knockout cells strongly inhibited TEAD-responsive promoter activity (Figure 5F,G). Similarly, co-depletion of ZO-1 and talin using siRNAs also attenuated TEAD-responsive promoter activity (Appendix A). These data agree with the reduced YAP activity in ZO-1-depleted cells plated on soft ECM (Appendix A). Thus, focal adhesions regulate the impact of ZO-1 on YAP activity.

### 3.6. YAP and GEF-H1 Drive Proliferation of ZO-1 Knockout Cells

We next asked whether YAP and GEF-H1 contribute to the effect of ZO-1 knockout on cell proliferation. Hence, we repeated the proliferation assay in combination with RNAi-mediated depletion of either YAP or GEF-H1. We included the depletion of ZONAB, a transcriptional and posttranscriptional regulator of gene expression that is also regulated by ZO-1 and GEF-H1 [4,5,34,44], as well as TAZ, a transcriptional activator related to YAP [13].

Figure 6A,B show that the depletion of YAP strongly inhibited cell proliferation, whereas the effect of TAZ depletion was much smaller and only significant in the ZO-1 single knockout cells. ZONAB depletion also attenuated the proliferation of both single and double knockout cells. GEF-H1 depletion reduced proliferation to levels of non-transfected controls cells, indicating that its induction played a central role in stimulating proliferation upon ZO-1 knockout.

As both transcription factors, YAP and ZONAB, are regulated by GEF-H1, we asked whether they influence each other (this paper and [44]). The depletion of either protein did not impact the expression levels of the other one (Figure 6C). We next employed luciferase reporter assays to measure impacts on activity. The activity of the TEAD-responsive promoter was not affected by ZONAB depletion. ZONAB activity was measured with a luciferase promoter construct that is repressed by ZONAB [4]. In control and YAP siRNA-transfected ZO-1 KO cells, ZONAB activity was similarly stimulated compared to wild-type MDCK cells as promoter activity was low but, as expected, strongly inhibited when directly depleted, which is indicated by increased promoter activity. YAP and ZONAB thus both contribute to ZO-1 knockout-induced proliferation but regulate transcription independently from each other.

### 3.7. ZO-3 Regulates GEF-H1 Expression and YAP Activity

ZO-1 associates with either one of its paralogs ZO-2 and ZO-3 [21,22,45,46]. Expression of truncated forms of ZO-3 have previously been linked to the stimulation of RhoA and actomyosin remodelling by unknown mechanisms [23,47]. Hence, we asked whether ZO-3 contributes to ZO-1-regulated YAP activation.

Knockout of ZO-1 alone resulted in a strong reduction in ZO-3 expression, which was comparable to levels in ZO-1/2 double knockouts (Figure 7A,B). Therefore, ZO-3, like GEF-H1 and TBK1, is impacted by ZO-1 knockout and may participate in the activation of downstream signalling mechanisms. To test this, we next depleted ZO-3 directly by transfecting siRNAs. Transfection of individual or pooled ZO-3 siRNAs efficiently depleted ZO-3 protein expression (Figure 7C). ZO-3 knockdown did not affect the junctional recruitment of occludin and ZO-1 (Figure 7D). However, all three ZO-3 knockdown conditions stimulated a nuclear accumulation of YAP and increased TEAD-responsive promoter activity (Figure 7F,G). Thus, ZO-3 depletion promoted the activation of YAP-regulated TEAD activity.

We next asked whether ZO-3 depletion was also sufficient for the activation of signalling intermediates. GEF-H1 was indeed induced by transfection of ZO-3 siRNAs (Figure 8A,B), and an increased phosphorylation of myosin regulatory light chain 2 (MLC2) was observed (Figure 8C,D). Both single (p-MLC2) and double (pp-MLC2) phosphorylated forms of MLC2 increased, indicating increased myosin activity. Immunofluorescence indicated increased myosin phosphorylation along the basal part of the cells (Figure 8E). As observed for ZO-1 knockout cells, phosphorylation of YAP was not affected, indicating that YAP regulation was Hippo-independent (Figure 8F,G). Phosphorylation of TBK1 was not stimulated by ZO-3 depletion (Figure 8F,H), indicating that depletion of ZO-3 was not sufficient to promote the activation of all signalling mechanisms stimulated by ZO-1 knockout and that the induction of GEF-H1 expression could occur without the activation of TBK1.

## 4. Discussion

We found that ZO-1 regulates a GEF-H1- and TBK1-dependent signalling network that activates YAP-regulated transcription and cell proliferation in a manner that depends on functional focal adhesions and is regulated by cell mechanics. ZO-1 depletion/knockout increased expression of GEF-H1 that, in turn, cooperated with the protein kinase TBK1 to stimulate the YAP/TAZ pathway in a Hippo-independent manner. Increased proliferation persisted in high-density cultures, indicating that ZO-1 is required for cell density-dependent proliferation control. GEF-H1 is known to also activate ZONAB to promote its activity in the regulation of transcription and translation [34,44]. ZO-3—a paralog of ZO-1 that selectively binds ZO-1 and not ZO-2—was expressed at reduced levels in ZO-1 knockouts, and its depletion was sufficient to induce GEF-H1 and Hippo-independent YAP activity. Therefore, GEF-H1 is a key target of ZO-1 signalling that stimulates independent transcriptional effector pathways based on YAP and ZONAB that regulate gene expression and drive cell proliferation.

Disruption of TJs by ZO-1/ZO-2 knockout selectively affected Hippo-independent YAP stimulation and did not impact YAP phosphorylation, indicating that the Hippo pathway did not regulate YAP downstream of ZO-1. The junctional complex contains multiple components of the Hippo pathway, many of which reside in the apical marginal zone [3,16]. Hence, the ZO-1 complex represents a distinct branch of junctional YAP regulation that acts via regulation of cell mechanics rather than regulating Hippo signalling. TJs are divided into a junctional core, containing proteins like ZO-1 and barrier-forming and -regulating proteins like claudins and occludin, and the apical marginal zone, containing apical polarity determinants and Hippo pathway components [3]. Thus, the two junctional subdomains also differ in how they regulate the YAP pathway as the ZO-1-dependent junctional core regulates YAP in a Hippo-independent but GEF-H1- and TBK1-dependent manner.

Observations on the role of YAP and cell proliferation by ZO proteins have been contradictory. ZO-1 was suggested not to regulate YAP and ZO-2, depending on the study, to either promote or inhibit YAP signalling and to impact Hippo signalling [12,19,48]. Similarly, experiments with stable knockdown cells of ZO-1 and ZO-2 suggested no effect on cell proliferation [12]. However, we detected activation of YAP/TEAD activity as well as increased proliferation. We could not detect an effect on YAP phosphorylation, indicating that reduced activity of the Hippo pathway was not involved in activation. While many of the previously reported experiments were conducted with MDCK cells such as ours, we have recently found that ZO-1 needs to be depleted efficiently or knocked out completely to induce clear loss-of-function phenotypes [8]; hence, the effects on cell proliferation may have been missed due to insufficient ZO-1 depletion. As ZO-1 and ZO-2 knockout increased apoptosis and necrosis, it is also possible that effects on proliferation were missed if cells were analysed under conditions that cause cell stress, such as serum starvation to synchronize cells. Similarly, ZO-1 knockdown in the adult retinal pigment epithelium in vivo, a nonproliferating epithelium, induces G1/S phase transition and cell death, whereas in other tissues, only death is reported [10,11]. Hence, the balance between cell proliferation and death in ZO-1-deficient cells may depend on other cell stressors acting on the epithelium studied. Increased proliferation was also not observed in a ZO-1 knockout mouse’s mammary epithelial Eph4 cells [12]. Unlike mammary epithelial cells in vivo, Eph4 cells do not express ZO-3. As ZO-3 is part of the YAP-regulating machinery downstream of ZO-1, its absence may reflect a permanent deregulation of that pathway in Eph4 cells.

Of the three ZO paralogs, ZO-3 has been studied the least. It was originally discovered as a protein co-precipitating with ZO-1 and was then later shown to possess a homologous structure to ZO-1 and ZO-2 [21,22,45,46]. Biochemical studies further revealed that ZO-2 and ZO-3 compete for binding to the same PDZ domain of ZO-1. Binding of ZO-3 to ZO-1 is important for efficient junctional recruitment of the latter as ZO-3 only shows weak punctate junctional localization in ZO-1 knockout cells [8]. Hence, downregulation of ZO-3 upon ZO-1 deletion is likely caused by the absence of junctional recruitment, which may stabilize the protein.

The function of ZO-3 is still poorly understood. In zebrafish, ZO-3 knockdown has been reported to affect the epidermal barrier function and, during gastrulation, to support ZO-1′s role in driving epiboly [49,50]. However, in mammals, the loss of ZO-3 in vitro or in vivo has not revealed defects in junction assembly [51]. A previous study involving cultured cells linked ZO-3 to the regulation of CDK4, a cell cycle kinase that is recruited to TJs [5], by recruiting and stabilizing the CDK4 regulator cyclin D1 to TJs [24]. Hence, ZO-3 may have proliferation promoting (i.e., stabilization of cyclinD1) and inhibiting (inhibition of GEF-H1 expression/YAP) activities. Expression of a truncated form of ZO-3 was shown to promote RhoA and actomyosin remodelling [23,47]. As ZO-3 can also bind to p120catenin, its effect on RhoA was attributed to p120catenin deregulation [23]. Our data now indicate that ZO-3 is, together with its interaction partner ZO-1, linked to the regulation of GEF-H1 expression, an activator of Rho GTPases.

In addition to GEF-H1, TBK1 is the other signalling protein we have identified here to regulate YAP activity downstream of ZO-1 ablation. Unlike GEF-H1 induction, ZO-3 does not affect TBK1 phosphorylation; hence, the activation of the kinase is not required for increased GEF-H1 expression. In lung fibroblasts, TBK1 stimulates Hippo-independent YAP/TAZ activation and is thought to be promoted by increased ECM stiffness [39]. However, GEF-H1 induction was not sufficient to promote TBK1 activity as ZO-3 knockdown increased GEF-H1 expression but not TBK1 activity. We have also not been able to identify a direct link between TBK1 and TJs using immunofluorescence or biochemical approaches. It is thus possible that indirect mechanisms link TBK1 activation to TJ disruption in epithelial cells. TBK1 can be activated by multiple stimuli and plays a central role in host defence mechanisms as well as in pathologies that disrupt normal protein homeostasis [52,53,54]. However, whether any of these signalling processes play a role in TBK1 regulation downstream of ZO-1 remains to be determined.

ZO-1 depletion induces cell-wide changes in cell mechanics, including increased traction on ECM, and the disruption of junction formation [8]. GEF-H1 plays a central role in this effect [8]. Hence, deregulation of GEF-H1 is a key process downstream of ZO-1 signalling that regulates cytoskeletal tension and thereby affects cell morphogenesis, focal adhesion formation, ECM traction, and gene expression. GEF-H1 is inhibited by TJ formation and redistributes upon knockout of ZO-1 [8,40]. Therefore, its activation is due to the release of junctional inhibition as well as increased expression.

Mechanotransduction at focal adhesions is an integral component of the TBK1- and GEF-H1-regulated signalling network activated by ZO-1 knockout. ECM stiffness modulated the impact of ZO-1 knockout on YAP activation, and the disruption of focal adhesion function by talin knockdown strongly inhibited YAP activation in knockout cells. Similarly, TBK1 knockdown (Figure 4) and GEF-H1 depletion or inhibiting mechanotransduction at focal adhesions (i.e., talin knockdown or very soft ECM) rescues junction formation upon knockdown/knockout of ZO-1 [8]. Thus, the ZO-1-regulated TBK1- and GEF-H1-dependent, mechanosensitive signalling network guides YAP activation and proliferation, as well as junctional integrity and, thereby, provides a functional regulatory link between TJs and ECM adhesion.

## 5. Conclusions

ZO-1 is a key structural component of TJs and is thought to regulate subcellular signalling mechanisms. Our data now indicate that ZO-1 is important for the cell density-dependent proliferation control as well as the regulation of death by necrosis and apoptosis. ZO-1 exerts its effect on cell proliferation by regulating the activity of two subcellular signalling proteins, GEF-H1 and TBK1, that in turn promote nuclear accumulation of the transcriptional co-activator YAP, promoting YAP/TEAD-regulated transcriptional activity. YAP as well as ZONAB, another transcription factor activated by GEF-H1 [44], then drive cell proliferation.

GEF-H1 is induced in the retinal pigment epithelium in response to epithelial mesenchymal transition stimulated by TGFβ signalling and in patients in responds to retinal insults [55]. Similarly, GEF-H1 induction is thought to be a key mechanism of cancer cell proliferation and was linked to escape from MAPK inhibition in cancer therapy [56,57,58,59]. Reduced ZO-1 expression and tight junction corruption has been reported in different types of cancer cells [9]; hence, GEF-H1 induction upon a reduced ZO-1 expression and the subsequent activation of transcriptional pathways like YAP/TAZ and ZONAB is likely to play an important role in the deregulation of cancer cell proliferation. Given the involvement of TBK1 in YAP regulation and the importance of TBK1 in cancer pathogenesis, it will be exciting to determine how the functional interplay between GEF-H1 and TBK1 impacts cancer cell behaviour and tumourigenesis.

## Figures and Tables

**Figure 1 cells-13-00640-f001:**
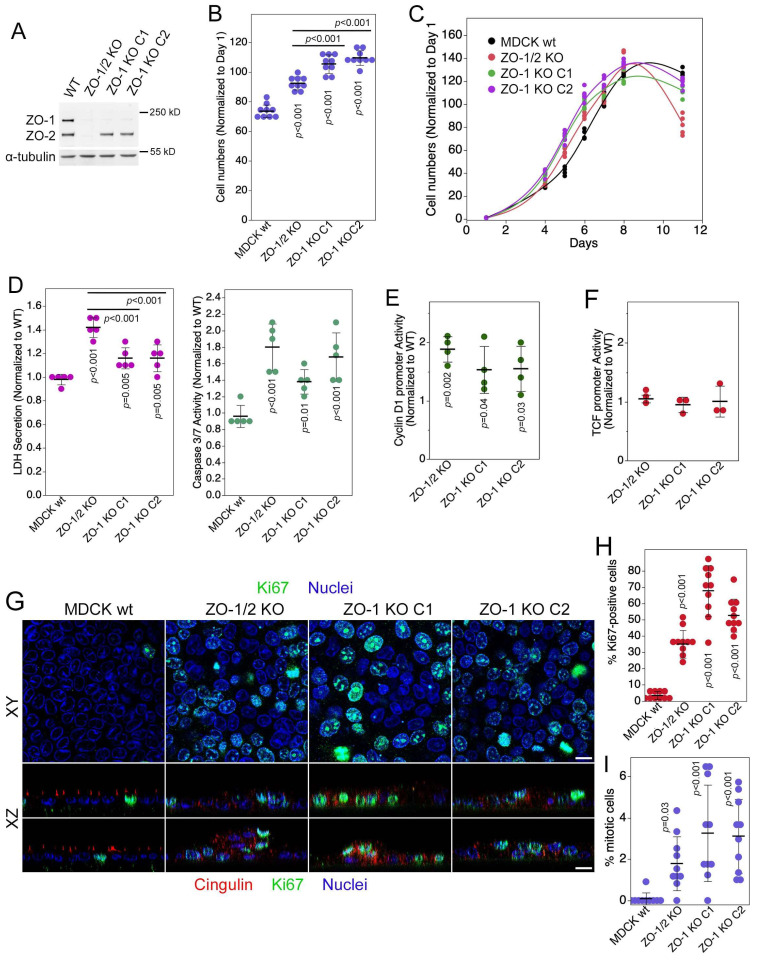
ZO-1 knockout induces cell proliferation and cell death. (**A**) Expression of ZO-1, ZO-2, and α-tubulin in control and knockout MDCK cells. (**B**) Cell proliferation was analysed in control and knockout MDCK cells by measuring cell numbers. Shown are values derived from day 6 of proliferation assays. (**C**) Control and knockout MDCK cells were grown over 11 days and cell numbers were analysed at the indicated timepoints. (**D**) Necrosis and apoptosis were measured in control and knockout MDCK cells as indicated. Shown are values derived from day 6 of proliferation assays. (**E**) Cyclin D1 promoter activity was measured by reporter gene assay. (**F**) Transcriptional activity of β-catenin/TCF was analysed by reporter gene assay. (**G**–**I**) Cells grown on filters for 6 days after confluence were immunostained for cingulin and Ki67, and nuclei were labelled with Hoechst dye. Confocal xy (**G**, top panel) and xz (**G**, lower panel) sections were acquired. The percentage of Ki67-positive cells (**H**) and mitotic indexes (**I**) were determined from xy sections. Quantifications show individual determinations, averages, standard deviations, and *p*-values derived from two-tailed *t*-tests comparing the knockout to wild-type cells. Magnification bars, 20 µm.

**Figure 2 cells-13-00640-f002:**
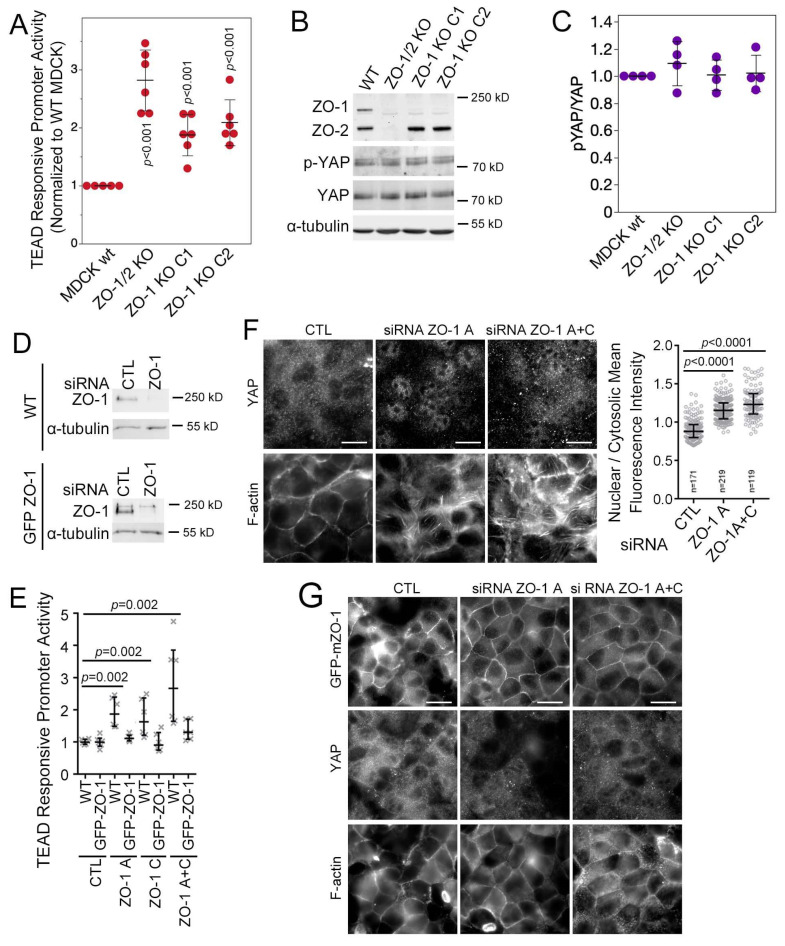
ZO-1 knockout stimulates YAP activation. (**A**) Wild-type (wt) and knockout MDCK cells were assayed for YAP/TEAD activity with a TEAD reporter gene assay. (**B**) Expression of ZO-1, ZO-2, YAP, α-tubulin, and levels of YAP phosphorylation were analysed by immunoblotting in wild-type and knockout MDCK cells. (**C**) Densitometry was used to quantify the ratio of phosphorylated YAP divided by total YAP as a measure for YAP phosphorylation. (**D**–**G**) Transfection of two different canine ZO-1-specific siRNAs (siRNA ZO-1 A and siRNA ZO-1 C) into wild-type and mouse GFP-tagged ZO-1 MDCK cells, followed by analysis through immunoblotting (**D**), TEAD reporter gene assay (**E**) or immunofluorescence with antibodies specific for YAP (**F**, wild-type MDCK cells; **G**, mouse GFP-ZO-1-expressing MDCK cells). The graph in panel F shows nuclear-to-cytosolic YAP measured in individual cells treated with ZO-1-specific siRNAs. Quantifications show individual determinations, averages, standard deviations, and *p*-values derived from two-tailed *t*-tests comparing the knockout to wild-type cells (**A**) or comparing control with ZO-1 siRNAs (**E**). The graph in panel (**F**) shows individual cells analysed, medians, interquartile ranges, and *p*-values derived from Wilcoxon tests. Magnification bars, 20 µm.

**Figure 3 cells-13-00640-f003:**
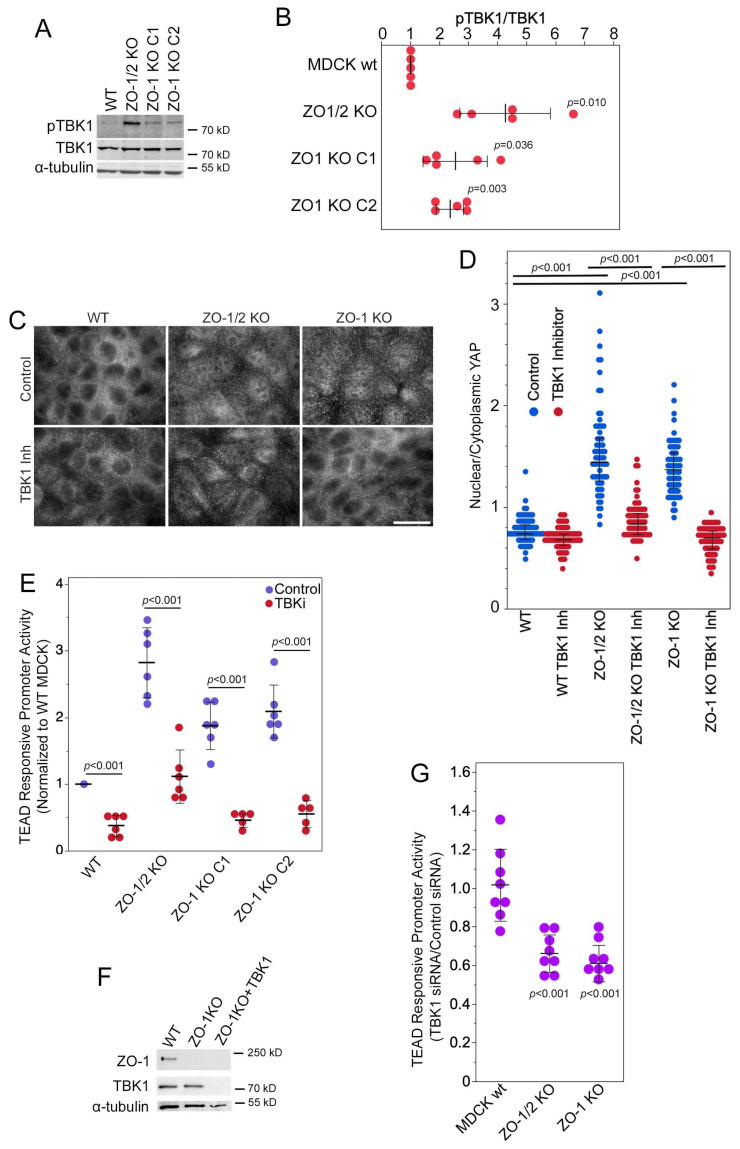
TBK1 promotes YAP activation. (**A**,**B**) Phosphorylation of TBK1 as a measure for its activation state was measured by immunoblotting. Immunoblots were analysed by densitometry followed by calculating ratios of phosphorylated/total TBK1. Values were normalized to wild-type MDCK cells. Shown are individual determinations, means, standard, and *p*-values derived from *t*-tests comparing to a test mean of 1. (**C**,**D**) Nuclear accumulation of YAP was visualized by immunofluorescence and quantified by measuring nuclear-to-cytoplasmic ratios. Shown are data points reflecting individual cells, medians, and interquartile ranges, and *p*-values derived from Steel–Dwass tests. (**E**) TEAD-responsive reporter gene assays were performed in control and knockout MDCK cells either treated with solvent or a TBK1 inhibitor. Shown are individual determinations, standard deviations, and *p*-values derived from *t*-tests comparing the indicated data pairs. (**F**) Knockdown of TBK1 in ZO-1 knockout MDCK cells was analysed by immunoblotting. (**G**) TEAD-responsive reporter gene assays were performed in control and knockout MDCK cells either treated with control or TBK1 siRNAs, and ratios were calculated. Shown are individual determinations, means, standard deviations, and *p*-values derived from *t*-tests comparing the indicated datasets to a test mean of 1. Magnification bar, 20 µm.

**Figure 4 cells-13-00640-f004:**
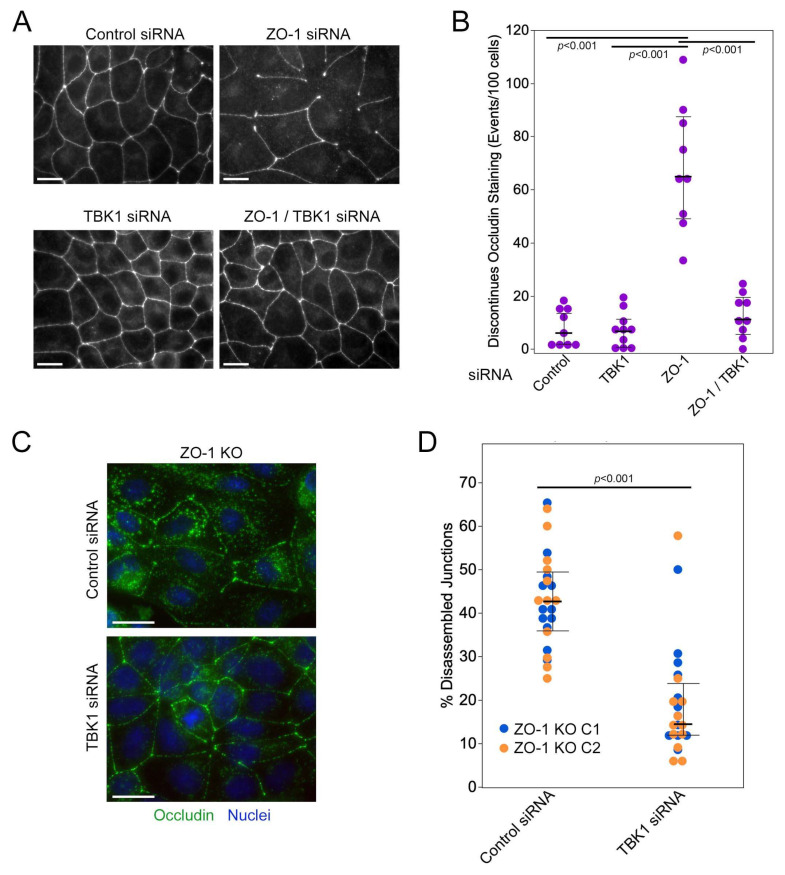
Depletion of TBK1 in ZO-1-deficient cells promotes TJ formation. (**A**,**B**) MDCK cells were transfected with siRNAs as in Figure 2G. The cells were then plated on Matrigel-coated coverslips and fixed processed for immunofluorescence after 2 days. Shown is staining for occludin. Panel (**B**) shows a quantification of TJ formation based on discontinuities in the occludin staining. Data points reflect images that were derived from two independent experiments (shown are also medians and interquartile ranges; *p*-values derived from Wilcoxon tests of the indicated data pairs). (**C**,**D**) ZO-1 KO cells were transfected with TBK1 siRNA and then plated, fixed, and processed as the cells in panel A. Shown is staining for occludin (green) and nuclei (blue). As occludin staining is weaker and discontinuous in ZO-1 KO cells, TJ formation was quantified by counting absent and present junctional segments. Shown are data derived from the two ZO-1 KO clones (blue dots, clone 1; orange dots, clone 2) and median and interquartile ranges for the pooled data (data points are derived from images collected from two independent experiments for each clone; *p*-values derived from Wilcoxon tests). Magnification bars, 20 µm.

**Figure 5 cells-13-00640-f005:**
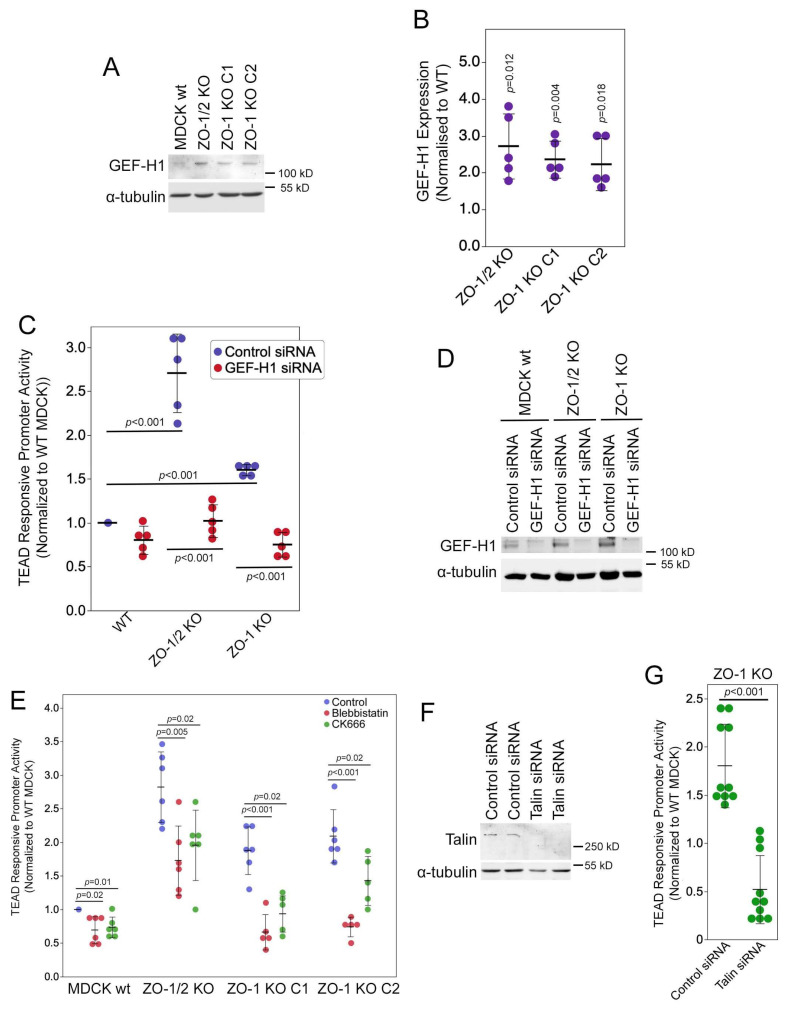
ZO-1 knockout induces GEF-H1 expression. (**A**,**B**) Wild-type and knockout MDCK cells were analysed by immunoblotting, and GEF-H1 expression was quantified using densitometry. (**C**,**D**) Cells were transfected with control or GEF-H1-targeting siRNAs and then analysed by TEAD reporter gene assay (**C**) or immunoblotting (**D**). (**E**) Control and knockout MDCK cells were treated for 16 h with inhibitors of myosin (Blebbistatin) or Arp2/3 (CK666) prior to analysis by TEAD reporter gene assay. (**F**,**G**) MDCK cells were transfected with control or talin targeting siRNAs and then analysed by immunoblotting (**F**) or TEAD reporter gene assay (**G**). Quantifications show individual determinations, averages, standard deviations, and *p*-values derived from two-tailed *t*-tests comparing the knockdowns to a theoretical mean of 1 (**A**) or the indicated data pairs (**C**,**E**,**G**).

**Figure 6 cells-13-00640-f006:**
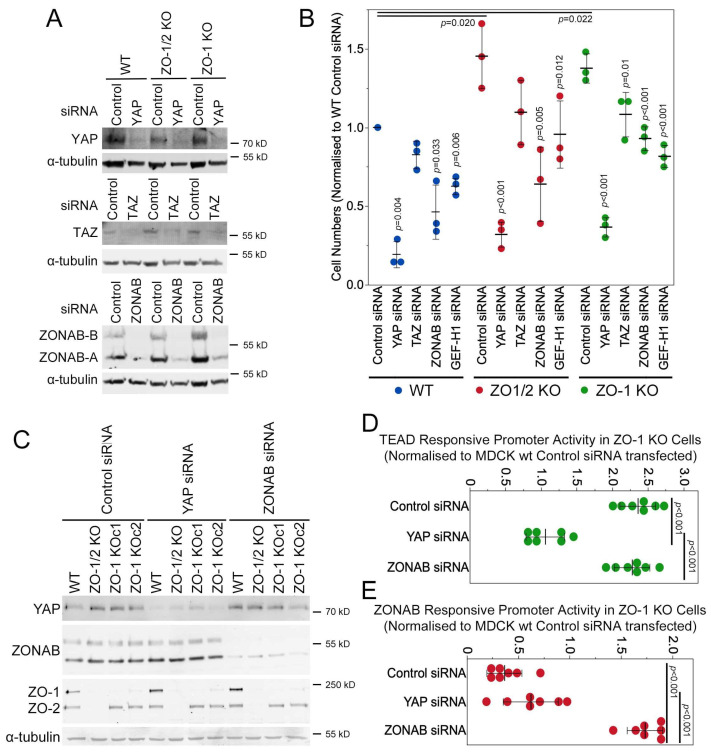
ZO-1 knockout-induced cell proliferation is YAP-, ZONAB-, and GEF-H1-dependent. Wild-type and knockout cell lines were transfected with the indicated siRNAs and then analysed by immunoblotting (**A**,**C**), by measuring cell numbers ((**B**), 6 days of proliferation), with a TEAD-responsive (**D**) or ZONAB-responsive (**E**) promoter assay. In panel B, values from wildtype cells are shown in blue, values from ZO-1/2 KO cells in red, and values from ZO-1 KO cells in green. Panels D and E show results obtained from ZO-1 KO cells normalized to control siRNA-transfected wild-type MDCK cells. Note that ZONAB functions as a transcriptional repressor in this assay; hence, reduced values indicate activation and increased values indicate the inhibition of ZONAB. Quantifications show individual determinations, averages, standard deviations, and *p*-values derived from two-tailed *t*-tests comparing the indicated pairs (if no paring is indicated, the *p*-value is for a comparison between the respective knockdown with control siRNA values of the same cell line).

**Figure 7 cells-13-00640-f007:**
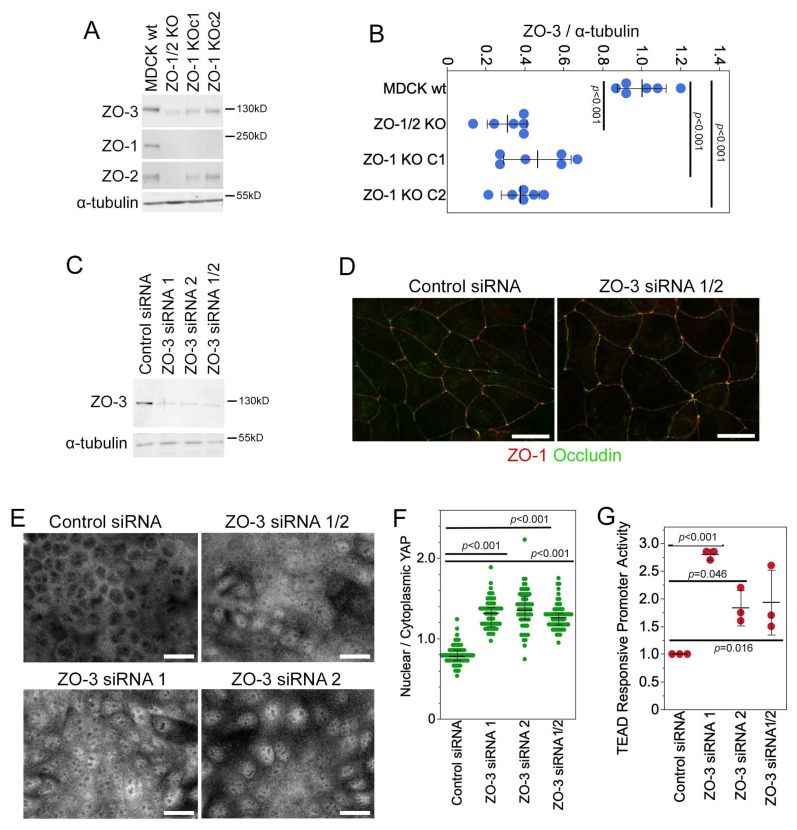
ZO-1 knockout reduces ZO-3 and ZO-3 depletion stimulates YAP. (**A**,**B**) Wild-type and knockout MDCK cells analysed by immunoblotting followed by densitometry for expression of ZO-3. Shown are individual determinations, means, standard deviations, and *p*-values derived from two-tailed *t*-tests comparing the knockout cell lines with wild-type cells. (**C**–**G**) MDCK cells were transfected with control or ZO-3-specific siRNAs as indicated and were then analysed by immunoblotting (**C**), immunofluorescence for the two TJ proteins ZO-1 and occludin (**D**), and immunofluorescence of YAP (**E**,**F**); quantification shows individual cells analysed, medians, interquartile ranges, and *p*-values from Wilcoxon tests, as well as the TEAD-responsive promoter assay (**G**); shown are individual determinations, means, standard deviations, and *p*-values from *t*-tests comparing knockdown to the control siRNAs. Magnification bars, 20 µm.

**Figure 8 cells-13-00640-f008:**
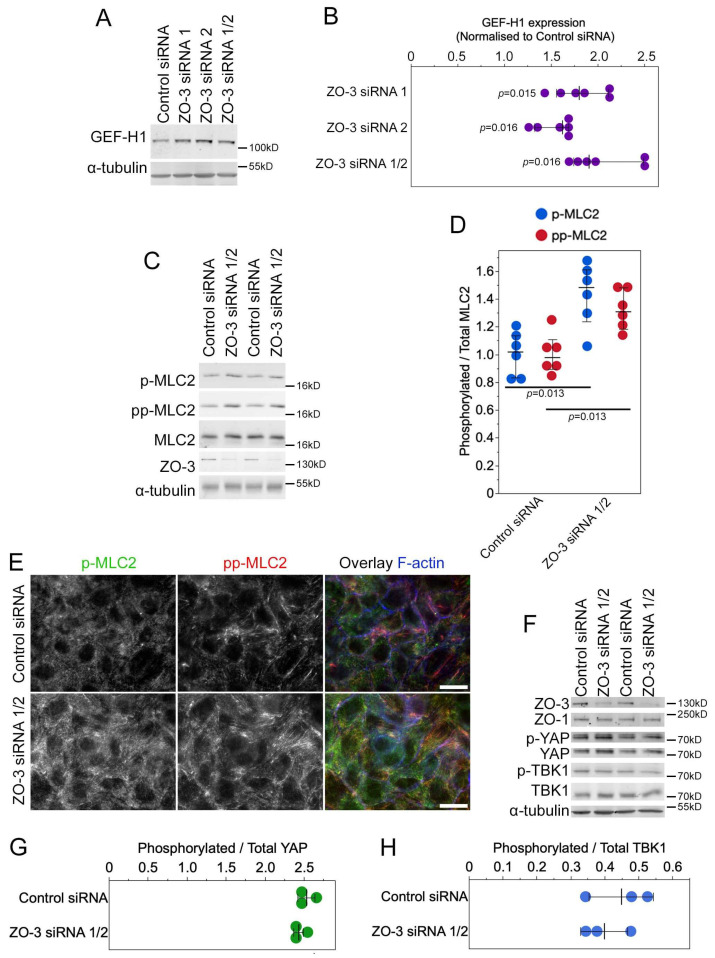
Depletion of ZO-3 stimulates GEF-H1 expression and phosphorylation of MLC2. MDCK cells were transfected with control or ZO-3 siRNAs (if not indicated, a pool of ZO-3 siRNAs was used) and then analysed by immunoblotting to determine GEF-H1 expression levels (**A**,**B**); shown are individual determinations, medians, interquartile ranges, and *p*-values derived from a signed-rank test, compared with a test median of 1, by immunoblotting to measure MLC2 phosphorylation (**C**,**D**); shown are individual determinations, means, standard deviations, and *p*-values derived from *t*-tests comparing knockdown to control siRNA datasets or immunofluorescence to visualize the distribution of phosphorylated MLC2 (**E**); shown are images from focal planes along the base of the cells showing the increase in basal actomyosin, or by immunoblotting as indicated (**F**). (**G**,**H**) Ratios of phosphorylated to total YAP (**G**) and TBK1 (**H**) were determined by densitometry of immunoblots. Shown are individual data points, means, and standard deviations. No statistically significant differences were observed by *t*-tests. Magnification bars, 20 µm.

## Data Availability

All data generated or analysed during this study are included in this published article and its Appendix A. Primary data are available from the communicating authors on request.

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
