# Peer review of "ZO-1 Regulates Hippo-Independent YAP Activity and Cell Proliferation via a GEF-H1- and TBK1-Regulated Signalling Network"

_cells, 2024, doi:10.3390/cells13070640_

Round 1

Reviewer 1 Report

Comments and Suggestions for Authors

The study by Haas and colleagues provides an extension of a previous work by the authors in which they further address molecular mechanisms through which the tight junction (TJ)-associated scaffolding proteins ZO-1 and ZO-2 regulate proliferation and morphogenesis of polarized epithelial cells. In this study, the authors observe an increased proliferative activity and an increase in cell death in cells lacking ZO-1. Additional depletion of ZO-2 decreases proliferation which is probably caused by an increase in cell death after ZO-2 depletion. As observed previously by the authors, the lack of ZO-1 also results in morphological alterations of the cells in a manner dependent on substrate stiffness suggesting an influence of ZO-1 on mechanical tension. The increased proliferation of ZO-1 KO cells is accompanied by increased nuclear translocation of YAP and increased YAP/TAZ transcriptional activity without changes in YAP phosphorylation, which suggests a Hippo pathway-independent mechanism of nuclear translocation of YAP. The authors identify TBK1 as regulator of YAP nuclear translocation in ZO-depleted cells. The authors also find that nuclear translocation of YAP in ZO-depleted cells is linked to the activity of GEF-H1 and regulation of actomyosin contractility, actin polymerization, and focal adhesion-mediated mechanotransduction. Finally, the authors find that ZO-3 is also involved in Hippo pathway-independent YAP-mediated TEAD activation, which as opposed to ZO-1 does not involve TBK1. The authors provide a careful and well controlled study on the role of ZO proteins in the regulation of epithelial cell proliferation. Even though partially overlapping with previous observations the study provides novel mechanistic insights in the molecular regulation of contact inhibition of proliferation by TJ components.  

Specific points

• In Fig. 1 the authors claim that ZO-1 KO cells reach a plateau at about day 8 of culture (Fig. 1C). However, all cell lines except for MDCK WT cells peak at day 8 but decline thereafter. What happens after day 11? Which of the two mechanisms – cell proliferation or cell death – prevails?   

• The occludin signals in Fig. 4C suggest a predominant cytoplasmic (perinuclear) localization of occludin both in control cells and TBK1 siRNA cells. The IF images do not reflect the statistical data shown in Fig. 4D. The IF images also do not match previous studies by the authors (e.g. Haas et al  2022 Cells 11, 3775). Better stainings should be provided.

• A key finding of the study is the identification of TBK1 as a ZO-1-regulated transcription factor. The activity of TBK1 is regulated by GEF-H1 (Ref 40). How do the authors envisage its regulation by ZO proteins? Is TBK1 associated with ZO proteins at the TJs? Is TBK1 localized at the TJs? Co-immunoprecipitation studies of TBK1 with ZO-1 and ZO-3 could provide some first insights. Also, immunfluorescence studies in polarized MDCK cells using ectopically expressed TBK1 constructs could provide some clues.

The observations i) that a depletion of ZO-1 results in strong reduction in ZO-3 levels (Fig. 7A), and b) that depletion of ZO-3 phenocopies several ZO-1 phenotypes (Nuclear YAP, GEF-H1 expression, MLC phosphorylation) suggests that the effects of ZO-1 depletion could be attributed to ZO-3 repression. How do the authors distinguish between ZO-1- and ZO-3-mediated effects? Re-expressing ZO-3 in ZO-1 KO cells could provide some clues.

• The pYAP and pTBK1 levels in ZO-3-depleted cells (Fig. 8) should be quantified.

Minor points

• Fig. 2F: please specify the abbreviations “siRNA ZO-1 A” and “siRNA ZO-1 C” in the figure legend (two different siRNAs against ZO-1?).

• Fig. 2B, F: the pYAP signals do not change in ZO-1/ZO2 KO lines suggesting that the Hippo pathway is not involved in the regulation. However, without changes in the levels of YAP phosphorylation the specificity of the antibody cannot be judged.  Please include a positive control sample for the pYAP antibody, e.g. subconfluent cells.

• Fig. 5B: the label “pAb” (“polyclonal antibody” ?) at the top of the data plot is unclear.

• Please refer to the corresponding figure after the statement “Inhibiting formin with …” (line 361).

• Fig. 7C - G: Please specify the “ZO-3 siRNA pool”. Is this a combination of siRNA 1 and siRNA 2? Please specify the siRNA used in Fig. 7D.

Reviewer 2 Report

Comments and Suggestions for Authors

In the manuscript, “ZO-1 Regulates Hippo-Independent YAP Activity and Cell Proliferation via a GEF-H1- and TBK1-regulated Signalling Network”, Haas et al study the role of tight junction (TJ) proteins ZO-1/2/3 in regulating cell proliferation via YAP/TAZ signaling independent of the Hippo signaling cascade. They show that the YAP translocates to the nucleus upon ZO-1/2 knockout without changing pYAP levels, indicating that this occurs independently of Hippo signaling. By using RNAi against GEF-H1, talin along with inhibitors of the actomyosin cytoskeleton, they show that changes in YAP localization are due to enhanced actomyosin contractility and traction forces exerted at focal adhesions.

The manuscript is very well organized and the study is designed very well. The text is well written and I found the manuscript a joy to read. I appreciate that the authors have provided validation of all the siRNA’s used in this study.

I have a few points of feedback which the authors should be able to address in short order:

1.        Cytoplasmic localization of YAP on glass/plastic:
YAP/TAZ have been shown to localize to the nucleus when cells are grown on stiff substrates (see for example, PMID 28951564). How is it that the authors find primarily cytoplasmic YAP in their cells (e.g. Fig 2F-G)? Is this unique to their cell type? This should be addressed in the text.

2.        pYAP detection

a.        In pYAP measurements reported by immunofluorescence in Fig 2, the pYAP signal appears to be completely cytoplasmic regardless of whether YAP is in the nucleus or cytoplasm. The antibody seems to work well for Westerns but I’m not so sure that it is picking up specific signal in immunofluorescence. The authors should include a positive control for pYAP in the supplement if they want to use the IF images to conclude a lack of change in YAP phosphorylation.

b.       In Fig 2, please quantify nuclear/cytoplasmic YAP ratio as was done in Fig S2. This is an important measurement accompanying the TEAD luciferase assay.

3.        Nuclear/cytoplasmic ratio of YAP signal
In the methods, the authors mention that they determine the best in-focus plane for their different channels. This is likely to be different for the nucleus, YAP signal, and actin. A better way to do this would be to use a maximum intensity projection of their z-stack and segment that into nucleus and cytoplasm.

4.        Cell proliferation and death assays
The methods section states that the samples were frozen at -20 for at least 18h before measuring proliferation or caspase3/7 levels. This seems very odd to me since this measurement should be done on living cells without any further manipulation. Freezing the cells at -20 is likely to do more damage to the cells. Please clarify.

5.        Small molecule inhibitors
The authors show that SMIFH2 has little to no effect on TEAD luciferase activity. In my experience, SMIFH2 is a very sensitive inhibitor and only works if reconstituted in fresh DMSO. The authors should make sure (and provide data in the supplement) that their inhibitors are working. They can stain for formins, Arp2/3, and MLC to ensure that their inhibitors are working in their cells.

6.        Model
The model here seems pretty clear where ZO-1 KO/KD results in release of GEF-H1 from TJ’s and subsequent activation. This in turn increases actomyosin contractility resulting in nuclear translocation of YAP/TAZ which causes transcriptional activation via TEAD.

Minor points:

1.        Abstract: The authors say, “We found that ZO-1 knockout increased cell proliferation, loss of cell density-dependent proliferation control, and promoted cell death.” How can increased cell proliferation be consistent with promotion of cell death? Please clarify.

2.        Fig 3D: y axis label is missing

3.        Fig 4A: Please show the nuclei as well. Does cell size also change upon ZO-1/2/3 KO?

4.        Fig 5 legend: Captions for panels D and E are mixed up

5.        Fig 6D-E: Which of the cell lines was used here (WT, KO, etc)? Please indicate in the figure

6.        Fig 7: Show that ZO-3 KD does not affect ZO-1 levels in the Western like has been done for 7A.

7.        Kindly provide the details of hydrogel preparation in this manuscript rather than citing earlier papers. This will keep this manuscript self contained.

Reviewer 3 Report

Comments and Suggestions for Authors

The authors indicate that ZO-1 controls cell proliferation and Hippo-independent YAP activity by activation of GEF-H1 and TBK1 which regulated mechanosensitive signaling network, by using MDCK ZO-1KO cells. Furthermore, ZO-1 KO increased loss of cell density-dependent proliferation control and promoted cell death (apoptosis). However, the mechanisms were unclear in the present study. This manuscript was well-written and new mechanisms for increase of cell proliferation by ZO-1 KO were described. The authors should more explain and discuss some phenomenon.

Specific comments:

1.     The authors should add graphic abstract (Schema) for readers.

2.     The authors wrote that ZO-1 KO promoted cell death. Is this cell death apoptosis?

3.     The authors should more explain about Talin and ppMLC and more discuss them.

4.     ZO-1 KO increased loss of cell density-dependent proliferation control and promoted cell death (apoptosis). If it is possible, please add the hypothesis about the mechanisms in Discussion

5.     All images of western blotting bands were weak. If it is possible, please change them.
